# Humoral Immune Responses in Patients with Severe COVID-19: A Comparative Pilot Study between Individuals Infected by SARS-CoV-2 during the Wild-Type and the Delta Periods

**DOI:** 10.3390/microorganisms11092347

**Published:** 2023-09-20

**Authors:** Maria Sukhova, Maria Byazrova, Artem Mikhailov, Gaukhar Yusubalieva, Irina Maslova, Tatyana Belovezhets, Nikolay Chikaev, Ivan Vorobiev, Vladimir Baklaushev, Alexander Filatov

**Affiliations:** 1Laboratory of Immunochemistry, National Research Center Institute of Immunology, Federal Medical Biological Agency of Russia, 115522 Moscow, Russia; mary.sukhova13@gmail.com (M.S.); mbyazrova@list.ru (M.B.); artem.mihaylov.2001@mail.ru (A.M.); 2Department of Immunology, Faculty of Biology, Lomonosov Moscow State University, 119234 Moscow, Russia; 3Department of Immunology, Peoples’ Friendship University of Russia (RUDN University) of Ministry of Science and Higher Education of the Russian Federation, 117198 Moscow, Russia; 4Laboratory of Cell Technology, Federal Research and Clinical Center for Specialized Types of Medical Care and Medical Technologies of the FMBA of Russia, 115682 Moscow, Russia; gaukhar@gaukhar.org (G.Y.); serpoff@gmail.com (V.B.); 5Engelhardt Institute of Molecular Biology, Russian Academy of Sciences, 119991 Moscow, Russia; 6Clinical Hospital #85, Federal Medical Biological Agency of Russia, 115409 Moscow, Russia; dr.imaslova@gmail.com; 7Laboratory of Immunogenetics, Institute of Molecular and Cellular Biology, Siberian Branch of the Russian Academy of Sciences, 630090 Novosibirsk, Russia; belovezhec@mcb.nsc.ru (T.B.); na_chik@mcb.nsc.ru (N.C.); 8Laboratory of Mammalian Cell Bioengineering, Skryabin Institute of Bioengineering, Research Center of Biotechnology of the Russian Academy of Sciences, 117312 Moscow, Russia; ptichman@gmail.com

**Keywords:** SARS-CoV-2, COVID-19, variants of concern, virus neutralization

## Abstract

Since the onset of the COVID-19 pandemic, humanity has experienced the spread and circulation of several SARS-CoV-2 variants that differed in transmissibility, contagiousness, and the ability to escape from vaccine-induced neutralizing antibodies. However, issues related to the differences in the variant-specific immune responses remain insufficiently studied. The aim of this study was to compare the parameters of the humoral immune responses in two groups of patients with acute COVID-19 who were infected during the circulation period of the D614G and the Delta variants of SARS-CoV-2. Sera from 48 patients with acute COVID-19 were tested for SARS-CoV-2 binding and neutralizing antibodies using six assays. We found that serum samples from the D614G period demonstrated 3.9- and 1.6-fold increases in RBD- and spike-specific IgG binding with wild-type antigens compared with Delta variant antigens (*p* < 0.01). Cluster analysis showed the existence of two well-separated clusters. The first cluster mainly consisted of D614G-period patients and the second cluster predominantly included patients from the Delta period. The results thus obtained indicate that humoral immune responses in D614G- and Delta-specific infections can be characterized by variant-specific signatures. This can be taken into account when developing new variant-specific vaccines.

## 1. Introduction

During the COVID-19 pandemic, a wide range of aspects of immune response to SARS-CoV-2 was studied and by now, many features for achieving immunity against this virus have been uncovered. Both innate and T-cell immune responses have been shown to play important roles in protection against COVID-19; however, it was firmly established that virus-neutralizing antibodies serve as the most critical factor determining protection from symptomatic SARS-CoV-2 infection [1]. Accordingly, most of the studies during and in the wake of the COVID-19 pandemic have focused on the humoral immune response during the course of the disease and/or vaccination.

The specific humoral response against SARS-CoV-2 has been extensively studied in different scenarios. These include examining the immune response during or shortly after acute infection [2], as well as monitoring the immunity over the following months, upon reinfection or immunization with various vaccines and boosters [3,4,5]. Both homologous and heterologous vaccination regimens, hybrid vaccination after COVID-19, and breakthrough infection have been tested [6]. In these studies, much attention was paid to the study of the virus-binding and virus-neutralizing activity of sera against a variety of SARS-CoV-2 lineages, ranging from Wuhan-Hu-1 and D614G strains to the latest Omicron variants, such as Omicron BQ.1 and XBB. Numerous data have been collected on the differences between different variants of concern (VOCs) in terms of transmissibility, contagiousness, relative severity of the disease, and their ability to escape from vaccine-induced neutralizing antibodies [7].

Despite a large amount of accumulated data, limited attention has been paid to direct comparison of immunity after COVID-19 caused by the different variants of SARS-CoV-2 [8,9,10]. In order to fill this knowledge gap, we examined the levels of humoral responses induced by natural infection with SARS-CoV-2 variants causing the COVID-19 in Moscow, Russia. Since the beginning of the COVID-19 pandemic, Moscow has experienced several waves of COVID-19, and the dynamics of circulating SARS-CoV-2 genetic variants in the Moscow region have been closely monitored [11,12]. According to the whole-genome sequencing analyses dating back to May–June 2020, the B.1 variant of SARS-CoV-2 which bears a single D614G substitution in the spike protein was predominant in Moscow [13]. Subsequently, numerous VOCs appeared and began to spread; however, during the COVID-19 wave in October–November 2021, only the Delta (B.1.617.2) variant of SARS-CoV-2 was detected in Moscow [14].

The aim of this study was to compare the effects of acute SARS-CoV-2 infections during the D614G and the Delta waves on the humoral immune response. To address this goal, we investigated the cross-reactivity of the antibodies in these two groups of infected individuals against the antigens from the wild-type and Delta variant. Additionally, the Omicron BA.1 subvariant, which has a high number of mutations in the spike (S) protein was also included as a distant antigen in the study. Notably, as SARS-CoV-2 evolves and new variants emerge, there is a clear need to develop a new, robust and simple method for detecting virus-specific antibodies. In this study, we propose two new cell-based approaches evaluating spike-binding and virus-neutralization activities that can be easily adapted to new antigenic variants.

## 2. Materials and Methods

### 2.1. Patients

The study included patients who experienced acute SARS-CoV-2 infection in May–June 2020 (*n* = 27) or October–November 2021 (*n* = 21). All the examined individuals were hospitalized at the Federal Research Clinical Center of the Federal Medical-Biological Agency of Russia (FRCC) and were characterized by moderate or severe course of COVID-19. The inclusion criteria were as follows: a positive rt-qPCR test, presence of antibodies against SARS-CoV-2 nucleocapsid protein, no history of vaccination against SARS-CoV-2, and no history of previous self-reported COVID-19 infection.

### 2.2. Serum Samples

Serum samples were collected in the acute phase of the disease at a late time point (median 21 days, IQR 18–33) from the onset of the disease. Blood samples were collected into heparinized vacutainer tubes (Sarstedt, Nümbrecht, Germany #04.1927). Blood plasma samples were obtained by centrifugation, aliquoted into collection vials and stored at −70 °C. Samples were named according to the patient IDs and tested using six serological assays (Appendix A).

### 2.3. Recombinant Proteins

In-house production of recombinant RBD proteins (residues 319–541) was described earlier [15]. In brief, His-tagged RBD was expressed using the HEK293 cells and purified from cell culture supernatant using affinity chromatography on Ni-NTA agarose resin (Novagen, St. Louis, MO, USA). The RBD from the Wuhan-Hu-1 strain was designated as wild-type (WT) RBD and it matched the RBD from the D614G strain. RBD variants used in the study had the following substitutions: L452R, T478K (Delta); G339D, S371L, S373P, S375F, K417N, N440K, G446S, S477N, T478K, E484A, Q493R, G496S, Q498R, N501Y, Y505H (Omicron BA.1).

Extracellular domain of the human angiotensin-converting enzyme 2 (residues 319–541) was fused to an immunoglobulin G crystallizable fragment (ACE2-Fc) as described earlier [16]. The ACE2-Fc was stably expressed in a DHFR-negative Chinese hamster ovary (CHO) DG-44 cell line (Thermo Fischer Scientific, Waltham, MA, USA) and was purified from the conditioned-culture medium using chromatography on MabSelect SuRe column (Cytiva, Marlborough, MA, USA).

ACE2-Fc was fluorescently labeled using Alexa Fluor 488 NHS Ester Succinimidyl Ester (Thermo Fisher Scientific, A20000) followed by the removal of excess dye by buffer exchange on PD-10 desalting column (Cytiva, #17-0851-01). ACE2-Fc was conjugated to the horseradish peroxidase using HRP Conjugation Kit (Abcam, Cambridge, UK, #Ab102890) according to the manufacturer’s instructions.

### 2.4. ELISA

The levels of RBD-specific IgGs were determined using an in-house ELISA test [17]. In brief, 96-well high-binding ELISA plates (Greiner Bio-One, Kremsmünster, Austria) were coated overnight with 2 μg/mL of recombinant RBD in PBS. Plates were then washed three times and blocked with blocking buffer (Xema Co., Moscow, Russia) for 1 h at room temperature. Serum samples from patients with COVID-19 were 2-fold serially diluted from 1:20 to 1:12,500 in blocking buffer and added into the wells. Plates were then incubated with samples for 1 h at room temperature. After washing in PBS with 0.05% Tween 20, the plates were incubated for 1 h with rabbit anti-human IgG antibody (Jackson Immuno Research, West Grove, PA, USA, Cat# 309-005-008), thoroughly washed and incubated with goat ant-rabbit IgG antibodies conjugated with horseradish peroxidase (Bio-Rad, Hercules, CA, USA, #1721019) for 1 h. ELISA plates were washed 7 times and developed for 10 min with 100 μL of tetramethylbenzidine chromogen solution (Xema Co., Moscow, Russia). The reaction was stopped with 50 μL 1M H2SO4 and absorbance at 450 nm was read with an iMark microplate absorbance reader (Bio-Rad). Each sample was measured in triplicate. To determine the concentration of WT and Delta RBD-specific IgGs, a serial dilution of anti-SARS-CoV-2 RBD-specific human monoclonal antibody iB12 known to recognize both RBD variants equally well was included on each plate, a calibration curve was built and IgG levels were calculated (µg/mL) [15]. When determining the levels of BA.1-specific IgGs, we used high-titer serum as a standard and antibody levels were expressed as relative units (RU). Baseline values for 8 pre-pandemic healthy donor serum samples were derived from the cryopreserved samples collected in 2017–2018.

### 2.5. Membrane-Based ELISA (mELISA)

HEK293 cells were seeded at a density of 3.6 × 10^6^ cells/100 mm Petri dish (NEST). Next day, cells were transiently transfected with 30 µg of pCAGGS-SΔ19 expression plasmid encoding the spike protein of the D614G strain, Delta or Omicron BA.1 VOC. Spike variants had the following substitutions: T19R, G142D, Δ156–157, R158G, L452R, T478K, D614G, P681R, D950N (Delta); A67V, Δ69–70, T95I, G142D, Δ143–145, Δ211, L212I, ins214EPE, G339D, S371L, S373P, S375F, K417N, N440K, G446S, S477N, T478K, E484A, Q493R, G496S, Q498R, N501Y, Y505H, T547K, D614G, H655Y, N679K, P681H, N764K, D796Y, N856K, Q954H, N969K, L981F (Omicron BA.1). Transfection was performed by the calcium phosphate method [17].

Seventy-two hours after transfection, cells were harvested and lysed using a standard freeze–thaw protocol [18]. In brief, cells were washed twice with PBS, scraped, and pelleted at 300 g for 10 min. The pellet was resuspended in PBS containing 100 mM PMSF, and lysed in three freeze–thaw cycles. Lysates were clarified at 300 g for 10 min at 4 °C and membrane fraction was pelleted by centrifugation at 30,000 rpm for 90 min at 4 °C. The pellet was resuspended in PBS and protein concentration was determined by measuring the absorbance at 280 nm. ELISA plates (Greiner Bio-One, Kremsmünster, Austria) were coated with 20 μg/well of membrane preparation. The reaction of serum samples with the spike protein in membrane preparations was developed and recorded in the same way as described above when performing RBD-specific ELISA. The antibody binding was measured in OD values. Wells coated with membranes derived from non-transfected HEK293 cells were used as negative controls. All samples were analyzed in duplicate.

### 2.6. Pseudotyped Virus-Neutralization Assay (pVNA)

Virus-neutralization activity in serum samples was determined using lentiviral particles pseudotyped with the SARS-CoV-2 S protein of the D614G strain or Delta and Omicron BA.1 VOCs. To produce SARS-CoV-2 S-pseudotyped virus-like particles (VLPs), HEK293 cells were co-transfected with three plasmids: lentiviral packaging plasmid pCMVΔ8.2R (Addgene, Teddington, UK), reporter plasmid pUCHR-GFP, and an expression plasmid pCAGGS-SΔ19 encoding the wild-type SARS-CoV-2 spike protein or those of the Delta or Omicron BA.1 VOCs [19].

Transfection was performed by the calcium phosphate method; 72 h after transfection, the supernatants were filtered through a 0.45 μm filter and concentrated 20-fold on Amicon^®^ Ultra-15 ultrafiltration cells with a 100 kDa cutoff (Merck, #UFC910008). Concentrated supernatants were then re-centrifuged at 30,000× *g*, 8 °C for 150 min. The pellet was resuspended in Opti-MEM medium. Before proceeding to pVNA, VLPs were titrated by limiting dilution with HEK293 cells stably transfected with a plasmid-expressing human ACE2 (HEK293-hACE2). A dose of viral particles which gave 50% green fluorescent protein (GFP)-positive cells was selected for use in the test. For pVNA, all plasma samples were heat-inactivated for 30 min at 56 °C prior to use. Serial dilutions of sera were pre-incubated with VLPs and then added to target cells and co-cultivated for four days. On the fourth day, the cells were re-suspended, and the percentage of GFP-positive cells was measured by flow cytometry. ID_50_ values were calculated using a normalized nonlinear regression with GraphPad Prism software, version 8.4.3. (Sigmoidal, 4PL).

### 2.7. Flow Cytometry-Based Surrogate Virus-Neutralization Assay (fcVNA)

This method measures the ability of sera to inhibit the binding of Alexa Fluor 488-labeled ACE2-Fc to HEK293 cells transiently expressing the S protein of interest. As described above, HEK293 cells were transiently transfected with pCAGGS-S∆19 plasmid encoding the S-protein of either the D614G strain, Delta or Omicron BA.1 VOCs.

The HEK293-spike cells (5 × 10^4^ cells/well) were mixed with an equal volume of serial serum dilutions (ranging from 1:2 to 1:64) and incubated for 1 h at room temperature. After washing, the cells were additionally incubated for 1 h with Alexa Fluor 488-labeled ACE2-Fc. Then, the wells were washed twice and the percentage of ACE2-positive cells was measured on a flow cytometer. Before the fcVNA assay, the transfection efficiency was monitored using staining with ACE2-Alexa Fluor 488. Preparations in which the percentage of S^+^ cells exceeded 75% were taken for fcVNA assay.

In the absence of serum, ACE2-Alexa Fluor 488 binding was the highest (set to 100%) and when the ACE2-Alexa Fluor 488 binding was completely inhibited, target cells did not produce a fluorescent signal. The results were presented as the serum dilution at which 50% inhibition (ID_50_) of cell binding was observed, calculated from the Sigmoidal titration plot, 5PL, in the GraphPad Prism program.

### 2.8. Surrogate Virus-Neutralization Assay (sVNA)

ELISA plates (Greiner Bio-One, Kremsmünster, Austria, #756071) were coated with 2 μg/mL of recombinant RBD diluted in PBS. After incubation overnight, the plates were washed with PBS, containing 0.025% Tween-20 (PBS-T) thrice and blocked with 200 µL/well of 5% BSA-PBS. Serum samples were diluted 1:20 and added to the plates to allow for binding of antibodies to the protein. The plates were incubated for 1 h. After washing, the plates were additionally incubated for 1 h with ACE2-Fc conjugated with horseradish peroxidase (50 ng/mL) [16]. ID_50_ values were calculated as described above.

### 2.9. Antibody-Dependent NK Cell Activation Assay (ADNKA)

Flat bottom 96-well plates (Greiner Bio-One, Kremsmünster, Austria) were coated with 200 μL 2 μg/mL of recombinant RBD (WT or Delta) in PBS overnight at 4 °C. Plates were blocked with 200 µL/well of 1% of bovine serum albumin with 5% sucrose in PBS for 1 h at RT. After five sequential washes with PBS, serum was added at a dilution of 1:40, incubated for 3 h at 37 °C, and washed 3 times with PBS.

Whole-blood samples from healthy donors were collected into heparinized vacutainer tubes (Sarstedt, #04.1927). PBMCs were isolated by density gradient centrifugation. NK cells were purified from PBMCs by negative selection using the Dynabeads Untouched human NK cells kit (Thermo Fisher Scientific, #11349D).

Immunomagnetically separated NK cells were resuspended in complete DMEM/F12 medium supplemented with 10% FBS (Cytiva, #SV30160.03), and plated at a density of 30,000 cells per well onto the RBD-coated plates in duplicate. After incubation for 6 h at 37 °C, Brefeldin A (5 mg/mL final concentration; Invitrogen, Waltham, MA, USA) was added and cells incubated for another 10 h. Activated NK cells were harvested, fixed, permeabilized using 0.1% saponin, and stained with PE/Cy5-labeled antibody for IFN-γ (Sony, Tokyo, Japan, clone 4S.B3).

Each ADNKA was performed with NK cells from a single donor. Cells were analyzed on a CytoFLEX S flow cytometer (Beckman Coulter, Brea, CA, USA). Up to 10 × 10^5^ cells were acquired per sample. Data were analyzed using FlowJo Software (version 10.6.1., Tree Star).

### 2.10. Statistical Analysis

All assays were carried out in duplicate, with the relevant positive and negative controls. The Kruskal–Wallis H test was used for comparison between multiple groups, and Dunn’s multiple comparisons test was performed using GraphPad Prism software, version 8.0.1. or Wilcoxon test for pairwise comparison. *p* < 0.05 was considered statistically significant. All statistical analyses were carried out using GraphPad Prism version 8.4.3 (GraphPad Software, San Diego, CA, USA). Heatmap generation and principal component analysis were performed with ClustVis using normalized data [20]. Data are presented as median values and interquartile ranges (IQR). Levels of statistical significance were denoted as * for *p* < 0.05, ** for *p* < 0.01, *** for *p* < 0.001, **** for *p* < 0.0001, ns for nonsignificant differences.

### 2.11. Ethics Statement

Written informed consent was obtained from each of the study participants before performing any study procedures. Study protocol was reviewed and approved by the Medical Ethical Committee of FRCC (#4-2020 28 April 2020) and conforms to the ethical guidelines of the 1975 Declaration of Helsinki.

## 3. Results

### 3.1. Study Design

Between May 2020 and April 2023, Moscow experienced at least six waves of COVID-19 cases (Figure 1A). Whole-genome sequencing analysis data (available on the GISAID server https://www.gisaid.org, accessed on 10 July 2023) showed a succession of different SARS-CoV-2 genetic variants (Figure 1B). At the early stage of the pandemic, SARS-CoV-2 evolved rather slowly [21] and throughout most of 2020 the B.1 variant was dominant. The B.1 spike protein differed from the Wuhan-Hu-1 isolate only by a single D614G mutation. At the turn of 2020 and 2021, the Alpha appeared briefly and the Beta VOC was also minimally present. At the beginning of 2022, all previous variants were completely replaced by the Delta (B.1.617.2) VOC. Various subvariants of Omicron have been establishing themselves in Russia since early 2022.

Our study included two groups of patients hospitalized for acute COVID-19. We obtained serum samples from patients who had been infected in May–June 2020 (*n* = 27) or in October–November 2021 (*n* = 21). We did not determine the genotypes of the infecting variants in patients; however, it can be stated with a high degree of certainty that individuals from the May–June 2020 group were infected with D614G strain, and those from the November–December 2021 group were infected with the Delta variant. Accordingly, the groups were classified as “D614G period” and “Delta period”.

The demographic characteristics of the patients are summarized in Appendix A. The two groups of patients were overall similar in terms of age, male/female ratio, and disease severity. The median age was 68 years (IQR 60–72) and 73 years (IQR 63–82) for the D614G-period and Delta-period patients, correspondingly. The study included 18 patients with the moderate form (non-ICU) and 30 patients with the severe (ICU) form of COVID-19. Samples were collected during the acute phase of the disease; the median time between the onset of symptoms and sample collection was 21 days (IQR 18–33).

### 3.2. RBD- and Spike-Specific IgG Response

Since the RBD region plays a critical role in mediating the interaction between the S protein of the coronavirus and the ACE2 receptor, RBD is the main target for neutralizing antibodies. First of all, we determined the levels of RBD-specific IgGs in serum samples. In a standard ELISA, we used WT and Delta RBDs, which fully corresponded to the antigens in the D614G and Delta periods. For comparative purposes, BA.1 RBD was also included in the analysis. IgGs from the D614G and Delta periods bound well to WT and Delta RBD, but much worse to BA.1 RBD (*p* < 0.0001; Appendix A).

Then, we compared anti-RBD IgG levels between the patients from the D614G and Delta periods and studied the cross-reactive activity of antibodies in these two groups. The activity of sera from the D614G period against WT RBD was 3.8 times higher than the cross-reactivity of sera from the Delta period (*p* < 0.01; Figure 2A, left panel). In contrast, the cross-reactivity of sera from the D614G period sera did not differ from the activity of the Delta sera against Delta RBD (*p* = 0.1161; Figure 2A, middle panel). Sera from either period reacted equally poorly with Omicron RBD (D614G vs. Delta period, *p* = 0.9909; Figure 2A, right panel).

The slight differences in cross-reactivity are probably due to the fact that WT and Delta RBDs differ from each other by only two amino acid substitutions. A more complete antigenic portrait of SARS-CoV-2 is presented on the transmembrane full-length S protein. S-specific antibodies were evaluated using an in-house developed mELISA assay. In this case, we used membrane preparations of HEK293 cells transiently transfected with a construct encoding a variant-specific S protein as an antigen. Similar to RBD-specific IgGs, the serum samples from the D614G and Delta periods showed strong binding to both WT and Delta S proteins, but a much weaker binding to the BA.1 S protein (*p* < 0.0001; Appendix A).

Then, we compared the levels of S-specific humoral responses induced by natural infection with SARS-CoV-2 during D614G and Delta periods. The anti-S IgG response in the D614G and Delta groups exhibited a comparable pattern to that observed for anti-RBD IgG (Figure 2B). The sera from the D614G period reacted more preferentially with the homologous WT spike than with heterologous Delta S protein (fold difference 1.96, *p* < 0.01; Figure 2B, left panel), while the Delta period sera were equally reactive towards the heterologous WT and homologous Delta S proteins (*p* = 0.2107; Figure 2B, middle panel). Some statistically significant differences were observed between the D614G- and Delta-period groups in terms of antibodies against the BA.1 S protein (*p* < 0.05; Figure 2B, right panel).

### 3.3. Virus-Neutralizing Response

To investigate the functionality of antibodies from D614G and Delta periods, we proceeded to evaluate the serum virus-neutralizing activity in pVNA. First, we compared how sera neutralize VLPs pseudotyped with different spike variants. Sera from the D614G period neutralized VLPs bearing the S protein of either WT or Delta variant approximately equally and 3.7 times more potently than the BA.1-pseudotyped VLPs (*p* < 0.0001; Appendix A). Sera from the Delta period neutralized VLPs pseudotyped with Delta S protein 3.5 and 3.9 times more effectively than WT- and BA.1-pseudotyped VLPs, respectively (*p* < 0.05, *p* < 0.01; Appendix A). No significant differences between serum samples from D614G and Delta period were observed in the level of virus neutralization in pVNA for WT, Delta, and BA.1 variants (Figure 3A).

We further characterized serum samples from the two groups of patients using a flow cytometry-based surrogate virus-neutralization assay (fcVNA) developed in this work. In this method, the S protein transiently expressed on HEK293 cells was used as an antigen and Alexa Fluor 488-conjugated ACE2-Fc fusion protein was used as the detection reagent. Neutralizing antibodies prevent the ACE2-Fc chimera from binding to the cell surface-expressed S protein. Representative flow plots for serum samples with and without neutralizing activity are shown in Appendix A. 

The levels of neutralization in fcVNA to each of the S protein (WT, Delta, and BA.1) variants tested were similar across both of the periods of infection (Figure 3B). fcVNA has an advantage over pVNA in terms of ease of setup, but is less sensitive than pVNA. Median values for ID_50_ in the fcVNA ranged from 2.3 to 10.1. Traditional sVNA, which is performed in the ELISA format, has a higher sensitivity. Recombinant RBD from WT, Delta and BA.1 variants was used as the plate-coating antigen and HRP-conjugated ACE2-Fc fusion was used as the detection reagent. Median values for ID_50_ in sVNA ranged from 21.7 to 66.5. Despite the higher sensitivity of sVNA, we were unable to detect differences in virus neutralization between the samples from the D614G and Delta periods (Figure 3C). The sensitivity of the Omicron variant to neutralization by serum samples from infected patients was very low.

### 3.4. Antibody-Dependent NK Cell Activation Assay (ADNKA)

RBD-specific antibodies, in addition to virus-neutralizing activity, can also have an effect or function associated with NK-cell–mediated antibody-dependent cell-mediated cytotoxicity, thereby contributing to a functional antiviral response [22,23]. To assess the effect of RBD-specific antibodies on NK cell activation, freshly purified NK cells isolated from healthy donor PBMCs were cultured in RBD-coated plates in the presence of serum samples from patients infected during the D614G and Delta periods. We observed RBD-specific Fc-dependent activation of NK cells which was accompanied by the expression of IFN-γ (Appendix A). After incubation with serum samples from healthy controls, no more than 1% of NK cells expressed IFN-γ, and this level was established as a baseline.

Functional RBD-specific antibodies that can induce NK cell activation were found in some serum samples of patients with COVID-19. Of the D614G- and Delta-serum samples, 37% (10/27) and 52% (11/21) had IFN-γ+ NK cells above the baseline when tested against WT RBD, correspondingly, whereas 89% (24/27) and 86% (18/21) of samples exceeded the baseline in ADNKA against the Delta RBD, respectively (Figure 4). The comparison of serum samples from the D614G and Delta periods showed that the levels of ADNKA for these groups did not differ significantly (*p* = 0.12 and *p* = 0.96, respectively).

### 3.5. Hierarchical Cluster and Principal Component Analyses

Next, we compared the samples from the D614G- and Delta-period groups for individual parameters obtained by various assays, i.e., we set out to examine the full complement of inter-relationships between all the variables. This task was completed by using hierarchical cluster and principal component analyses, which included 45 samples measured by 16 humoral immune response parameters.

The constructed dendrograms indicated the existence of two well-separated clusters (Figure 5A). The first cluster mainly consisted of D614G-period patients (total *n* = 24, D614G and Delta periods *n* = 20 and 4, respectively) and the second cluster predominantly included patients from the Delta period (total *n* = 21, D614G and Delta periods *n* = 6 and 15, respectively). When comparing the samples using principal component analysis, we did not observe well-separated clusters. However, it could be noted that serum samples from the Delta period had significantly greater variability than those from the D614G period. The serum samples from the D614G-period cluster were quite compact and were completely included within the Delta-period cluster.

## 4. Discussion

The vast majority of currently approved COVID-19 vaccines are based on the reference viral strain originally identified in Wuhan. Initially, the developed vaccines successfully protected against COVID-19. However, due to the continuous evolution of SARS-CoV-2 and the emergence of new VOCs, antibodies generated in the vaccinated individuals displayed progressively reduced virus-neutralizing activity against antigenically distinct variants, in particular, Delta and especially against Omicron and its sublineages [24]. Since the protective efficacy of the original vaccines against VOCs has declined, this prompts the development of variant-adapted vaccines. Several variant-adapted vaccines have been developed and approved, for which increased immunogenicity and protection against Omicron-related subvariants have been reported [25]; however, the relative benefits of using variant-adapted vaccines remain controversial [26]. The ambiguity in the use of variant-specific vaccines is compounded by the fact that the development and trials of new vaccines is laborious, complex and expensive. In addition to vaccination, protection against COVID-19 can be achieved by treating with monoclonal antibodies (mAbs) against SARS-CoV-2 antigens. The problem of increased immune escape by the new SARS-CoV-2 variants is common to both variant-specific vaccines and therapeutic mAbs [27], so the study of the cross-reactivity of sera from recovered patients with different SARS-CoV-2 variants can also serve as a valuable resource for the development of novel prophylactic monoclonal antibodies (mAbs).

For variant-specific vaccines, the critical question is whether immunization can induce cross-reactive (cross-neutralizing) antibodies to other variants. This can be easily assessed using animal models [28]. It was shown that variant-specific immunization caused higher activity of neutralizing antibodies against the corresponding variant than against the original strain [10].

An alternative approach to evaluate variant-specific immunization is to compare serum samples from patients who had a prior infection with different SARS-CoV-2 variants in terms of their cross-reactivity. However, only a few detailed reports on this subject have been published [8,9]. In particular, it was demonstrated that Omicron infections lead to increased antibody binding to pre-Omicron S variants [9]; at the same time, sera from the Delta and pre-Delta periods were similar in binding to S protein variants. In order to compare more comprehensively the humoral immunity in patients from the D614G and Delta periods, we used a set of methods, both standard (ELISA, pVNA, and sVNA) and especially designed for this case (fcVNA and mELISA).

Capture antigens used in ELISA should mimic natural antigenic epitopes as much as possible. The S protein is a large transmembrane, heavily glycosylated polypeptide with molecular weight 180–200 kDa. Human HEK293 or Chinese hamster ovary (CHO) cells represent the best option for the production of a soluble extracellular domain of the S protein. However, due to the large size of the S protein, it belongs to a class of difficult-to-express proteins [29,30]. The antigenic portrait of the S protein is most complete when it is present in a prefusion trimerized conformation. However, this conformation is metastable. The prefusion state of the SARS-CoV-2 S trimer protein can be stabilized by the addition of two or six proline residues (S-2P or HexaPro variant) and abolishing the furin cleavage site [31,32,33]. In addition, depending on the position of the RBD domains, the S trimer can be in several states: closed, partially, or completely open conformation.

When switching to a new variant of the S protein, the production and purification procedure must be adjusted anew. Production of a set of mutant variants of the S protein in its native conformation is a very labor-intensive and expensive process. Thus, the search for alternative ways to obtain S proteins is an urgent task. In this study, we proposed the mELISA method, in which the full-length S protein in the membrane fraction of transfected cells is used as an antigen immobilized on microplates. In this method, the S protein is presented in a native conformation in a tagless form, and its production is very simple and easily adaptable to new mutant variants. The full-length transmembrane S protein from transiently transfected cells possesses the complete antigenic spectrum and in combination with mELISA could represent a rapid production platform for detecting antibodies against mutant variants. The idea to use the transmembrane S protein as ligand instead of isolated recombinant antigen was also used by us in the flow cytometry-based surrogate virus-neutralization assay (fcVNA).

The binding of recombinant ACE2 with the membrane-bound S protein is more consistent with the interaction between the RBD of SARS-CoV-2 with the ACE2, a cell surface receptor of the human host cell, than when a purified recombinant RBD, which is commonly used in sVNA [34], is considered. Since a single, invariant reagent, namely, ACE2-Fc, is used as a labeled probe for various S protein variants, the method can be easily adapted to any S variant. Finally, the protocol of fcVNA allows experiments to be performed on the full-length S protein at a minimal cost. It should be noted that traditional pVNT is more sensitive than sVNT and fcVNA.

For the examined patients, we did not perform genotyping of the SARS-CoV-2 variants they were infected with; however, the time of collection was chosen in such a way as to ensure the variant-specificity of the obtained samples to the greatest extent. Taking into account the published data on SARS-CoV-2 sequencing carried out in Russia, it can be argued that the examined groups represent fairly pure populations that were infected with either D614G or Delta variants. During the first wave of 2020, there were no vaccinated or reinfected patients, and in the wave of 2021, their numbers were still very small and easily separable from the general sample. It would have been of great interest to include patients from the Omicron period in this study; however, subsequent waves were fueled by several Omicron subvariants. In addition, by this time, the percentage of vaccinated people or those infected with previous variants had noticeably increased, and a significant proportion of the population already had some level of immunoreactivity against SARS-CoV-2.

The aim of this study was to determine whether variant-specific infections have a unique molecular signature. Our comparison of patients from the D614G and Delta periods in terms of individual parameters of the humoral response revealed only minimal differences between these groups. This is not surprising, because compared to the D614G strain, the Delta S variant has only nine amino acid substitutions, including two deletions in the S protein and these antigens are more than 98% identical.

However, we found that serum samples from the D614G period reacted with RBD and S from the WT strain 3.9 and 1.6 times more strongly than with the corresponding antigens from the Delta variant. Interestingly, serum samples from the Delta period demonstrated similar levels of IgGs against the WT and Delta variants of RBD and S. This demonstrates that Delta infection is a good inducer of cross-reactive antibodies against a predecessor WT strain. The Omicron infection has recently been found to induce cross-reactive antibodies against heterologous spike variants [9]. We hypothesize that the Delta infection similarly contributed to increased antibody binding to the D614G spike variant.

A more thorough analysis using the antigenic maps showed that the serum samples from D614G and Delta patients were located at a small but distinguishable antigenic distance from each other, and formed distinct clusters [8]. In line with these data, our cluster analysis also showed some differences between the studied groups of patients. Thus, taking into account several humoral parameters, it was possible to identify the signatures of humoral immunity in the D614G- and Delta-period groups. The presented study can be considered as a pilot study. In order to evaluate variant-specific humoral responses more accurately, additional epidemiological studies will be needed.

## 5. Limitations of the Study

The limitation of this study is the relatively small sample size. In addition, the assignment of the patients to either D614G- or Delta-infected groups was based on the period of disease, but was not directly confirmed by genomic sequencing.

## 6. Conclusions

The results of our study show that humoral immune responses in D614G- and Delta-specific infections can be characterized by variant-specific signatures. These data can be taken into account when developing new variant-specific vaccines.

## Figures and Tables

**Figure 1 microorganisms-11-02347-f001:**
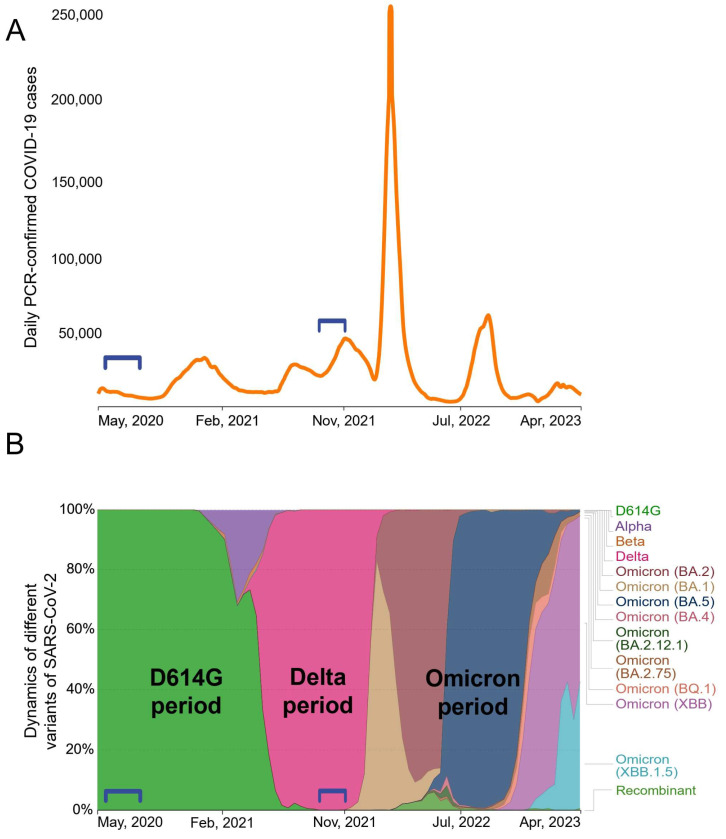
The dynamics of the spread of SARS-CoV-2 variants in Moscow population during the period from May 2020 to April 2023. Numbers of confirmed COVID-19 cases (**A**). Relative prevalence of SARS-CoV2 VOCs. Brackets denote the two periods of infection, D614G and Delta, in the study groups. (**B**) showed a succession of different SARS-CoV-2 genetic variants.

**Figure 2 microorganisms-11-02347-f002:**
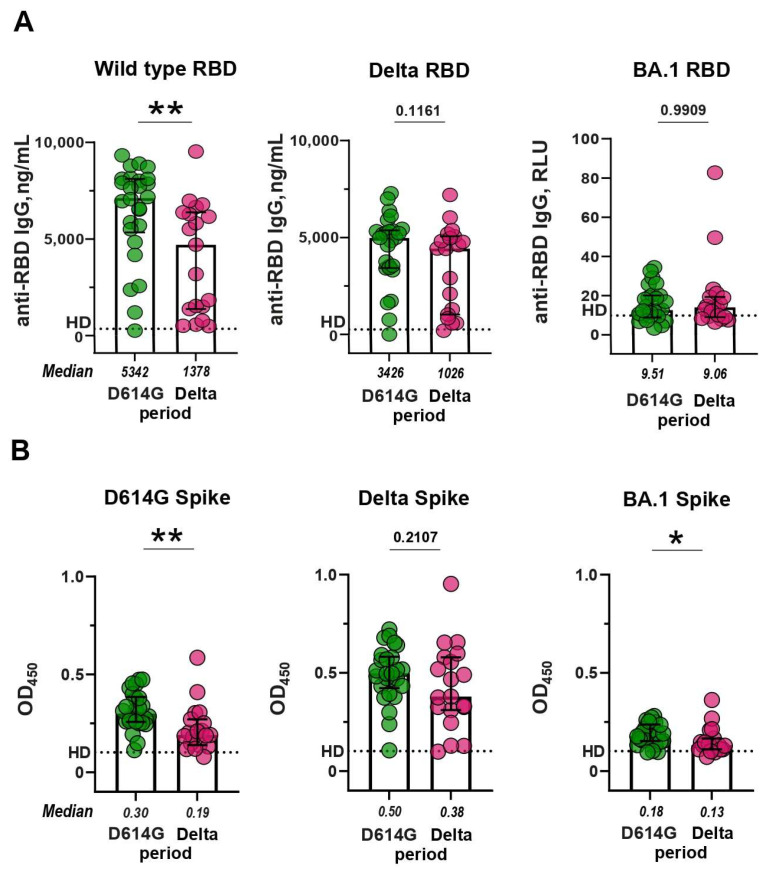
RBD- and S-binding activity of sera from patients with COVID-19 infected during the D614G and Delta periods. (**A**) Levels of serum IgG against WT, Delta, and BA.1 RBD, measured by ELISA. (**B**) Levels of serum IgG against WT, Delta, and BA.1 S protein, measured by mELISA. Dotted lines indicate the cut-off value for differentiating a positive response from a background response in the pre-pandemic samples from healthy donors (HD). ** *p* < 0.01, * *p* < 0.05.

**Figure 3 microorganisms-11-02347-f003:**
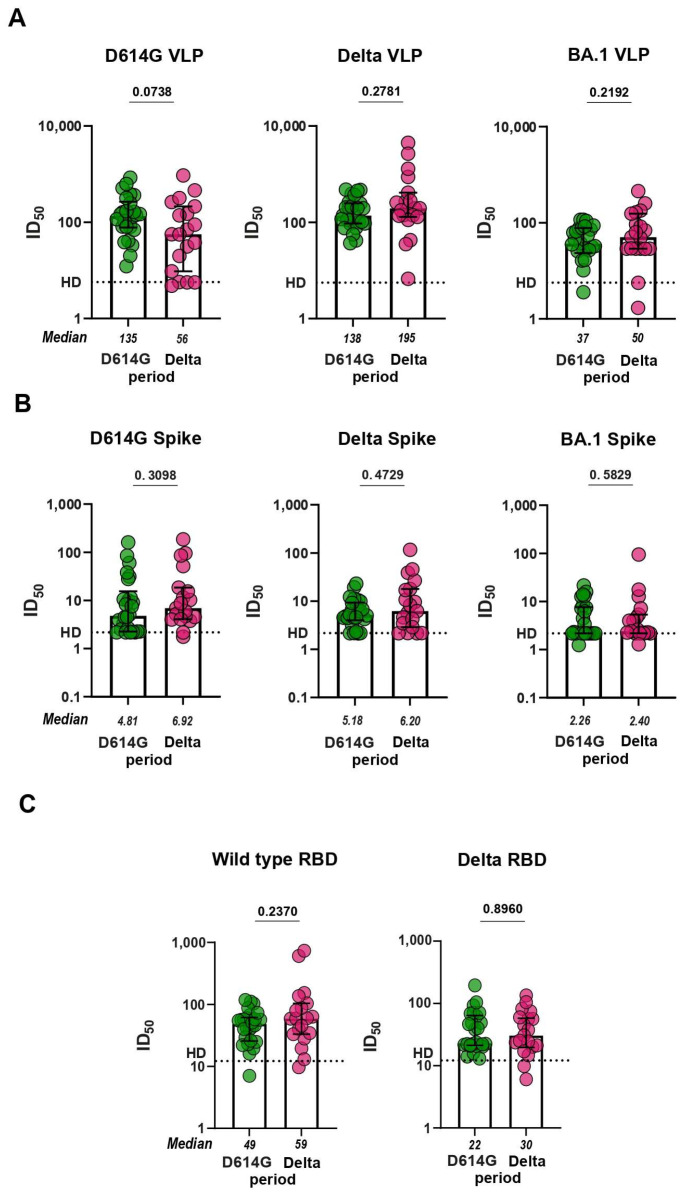
Neutralization antibody titers (ID_50_ values) against SARS-CoV-2 variants for sera from COVID-19 patients infected in the D614G and Delta period. (**A**) Neutralization of VLPs pseudotyped with the S protein from WT, Delta, and BA.1 Omicron variants. (**B**) Antibody-mediated blocking of ACE2-Alexa488 binding to HEK293 cells transiently transfected with S protein from WT, Delta, and BA.1 Omicron variants evaluated in fcVNA. (**C**) Antibody-mediated blocking of ACE2-HRP binding with RBD from WT, Delta, and BA.1 Omicron variants evaluated in sVNA.

**Figure 4 microorganisms-11-02347-f004:**
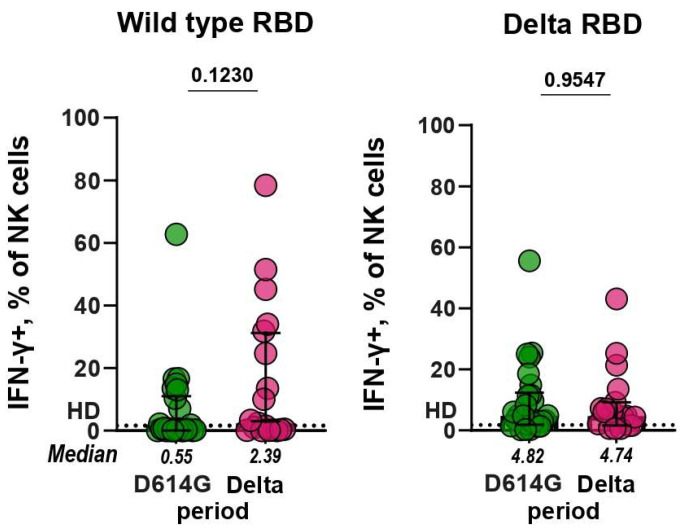
IFN-γ expression by NK cells in ADNKA. Plates were coated with wild-type (**left panel**) or Delta (**right panel**) RBD. NK cells were cultivated on plates in the presence of serum samples (dilution 1:40) from patients infected during D614G and Delta periods.

**Figure 5 microorganisms-11-02347-f005:**
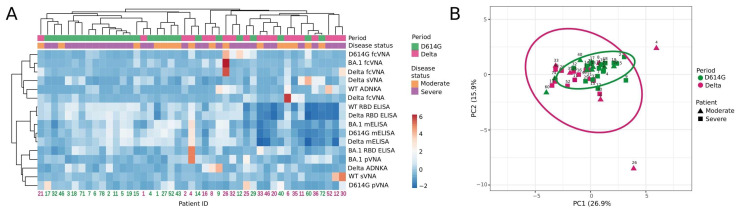
Hierarchical cluster (**A**) and principal component analysis (**B**) of serum samples from patients hospitalized with COVID-19 during D614G and Delta period. (**A**) Columns denote patients with their IDs. Rows correspond to immune response variables. Dendrograms on the top illustrate the clustering of patients. Immune response measurement values are color-coded according to the key shown on the right. (**B**) Principal component analysis of patients with COVID-19. Patient IDs are shown. D614G- and Delta-period clusters are indicated by the ovals. Results are shown for individual samples (symbols) from D614G period (*n* = 26) and Delta period (*n* = 19) patients.

## Data Availability

The data that support the findings of this study are available from the corresponding author upon reasonable request.

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
