# Peer review of "Humoral Immune Responses in Patients with Severe COVID-19: A Comparative Pilot Study between Individuals Infected by SARS-CoV-2 during the Wild-Type and the Delta Periods"

_microorganisms, 2023, doi:10.3390/microorganisms11092347_

Round 1

Reviewer 1 Report

The authors intended to compare the parameters of the humoral immune responses in two groups of patients with acute COVID-19 who were infected during the circulation period of the D614G and the Delta variants of SARS-CoV-2. Their results indicate that humoral immune responses in D614G- and Delta-specific infections can be characterized by variant-specific signatures.

Several suggestions

1.      Line 106, [human the] or [the human]?

2.      Line 117, please write down the full name for [ELISA].

3.      It is suggested to put [Supplementary Figure 1. Study design.] in [Materials and methods] section.

4.      In supplementary Figure 2B, is the p-value between Delta and BA.1 correct?

5.      In supplementary Figure 3A, please discuss why D614G period sera reacted better with Delta spike than that of WT.

6.      In supplementary Figure 4, please add [of D614G (А) and Delta (B) period serum samples] in the figure legend.

7.      In fcVNA assay, (line 183, HEK293 cells transiently expressing the S protein of interest), would transfection efficiency variations affect the results (e.g., Supplementary Figure 5)?

8.      In supplementary Figure 5B, please add [of D614G (А) and Delta (B) period serum samples] in the figure legend.

9.      Lines 476-477, [Omicron infection has recently been found to induce cross-reactive antibodies against heterologous Spike variants [9]], please discuss could Delta infection induce cross-reactive antibodies against WT?

10.  Except ADNKA, some of the other five assays could be the surrogate for the neutralization assay. Please discuss which one is the most suitable one.

Author Response

Author Response to Reviewer 1

We thank the Reviewer for their favorable review of our manuscript entitled “Humoral Immune Responses in Patients with Severe COVID-19: A Comparative Study between individuals infected by SARS-CoV-2 during the Wild-type and the Delta periods”. Our manuscript has been revised according to the suggestions and concerns raised by the Reviewer.

Below, please find our point-by-point responses to the Reviewers’ comments. Additionally, we specify the corresponding line numbers where the text has been added or modified and highlight these changes in green in the revised submission.

Point #1: Line 106, [human the] or [the human]?

Response: Corrected to “the human angiotensin-converting enzyme 2”.

Point #2: Line 117, please write down the full name for [ELISA].

Response: Corrected to “Enzyme-linked immunosorbent assay (ELISA)”.

Point #3: It is suggested to put [Supplementary Figure 1. Study design.] in [Materials and methods] section.

Response: The reference to Supplementary Figure 1 (Study design) has been moved to the “Materials and Methods” section (Line 99).

Point #4: In supplementary Figure 2B, is the p-value between Delta and BA.1 correct?

Response: Yes, it is correct. The Delta samples had a very large spread, which is why the p value is so high.

Point #5: In supplementary Figure 3A, please discuss why D614G period sera reacted better with Delta spike than that of WT.

Response: We thank the Reviewer for a very insightful question. In the mELISA test, it is rather difficult to compare responses to different antigens. Antigens can differ in efficiency of transfection and subsequent expression on HEK293 cells. It would be ideal to equalize or normalize the various antigens (D614G, Delta and BA.1 Spike) before coating of ELISA plates. Unfortunately, we do not have a monoclonal that would bind to all of these antigens equally and could be used for normalization. However, it is quite correct to compare sera from different periods on the same antigenic preparation. It is these results that were included in the main part of the results. Nevertheless, we considered it possible to present in the Supplementary the results of comparing responses to different antigens. We hope that the reviewer will treat these data with indulgence, especially since they have no effect on the conclusions drawn.

Point #6: In supplementary Figure 4, please add [of D614G (А) and Delta (B) period serum samples] in the figure legend.

Response: The figure legend was corrected.

Neutralization antibody titers (ID50 values) against VLP pseudotyped with D614G, Delta, and BA.1 Spike variants in pseudotyped virus neutralization assay (pVNA) of D614G (left) and Delta (right) period serum samples.

Point #7: In fcVNA assay, (line 183, HEK293 cells transiently expressing the S protein of interest), would transfection efficiency variations affect the results (e.g., Supplementary Figure 5)?

Response: We supplemented the text with the following phrase (Lines 193-195):

Before the fcVNA assay, the transfection efficiency was monitored using staining with ACE2-Alexa Fluor 488. Preparations in which the percentage of S+ cells exceeded 75% were taken for fcVNA assay.

Point #8: In supplementary Figure 5B, please add [of D614G (А) and Delta (B) period serum samples] in the figure legend.

Response: The figure legend was corrected.

B - neutralization antibody titers (ID50 values) against WT, Delta, and BA.1 Spike variants in fcVNA of D614G (left) and Delta (right) period serum samples.

Point #9: Lines 476-477, [Omicron infection has recently been found to induce cross-reactive antibodies against heterologous Spike variants [9]], please discuss could Delta infection induce cross-reactive antibodies against WT?

Response: As a discussion, we have added a phrase. Please see Lines 488-489.

This demonstrates that Delta infection is a good inducer of cross-reactive antibodies against a predecessor WT strain. Omicron infection has recently been found to induce cross-reactive antibodies against heterologous Spike variants [9]. We hypothesize that Delta infection similarly contributed to increased antibody binding to the D614G Spike variant.

Point #10: Except ADNKA, some of the other five assays could be the surrogate for the neutralization assay. Please discuss which one is the most suitable one.

Response: In Lines 461-462 we have added following phrase:

It should be noted that traditional pVNT is more sensitive than sVNT and fcVNA.

Reviewer 2 Report

I read with great interest the paper authored by Sukhova et al entitled "Humoral Immune Responses in Patients with Severe COVID-19: A Comparative Study between individuals infected by SARS-CoV-2 during the Wild-type and the Delta periods".

In my humble opinion, the manuscript is metodologically correct and the limitation of the study well explained. 

Author Response

Author Response to Reviewer 2

We thank Reviewer #2 for the favorable evaluation of our manuscript and we are pleased to receive such positive feedback.

Reviewer 3 Report

The manuscript represents an interesting serologic analysis to two different variants of SARS CoV-2 virus. The article is interesting, although the low number of patients limited the study. The specificity of the assays are validated, although controls with high or low antibody binding are missing.   The Fc cell activation of NK cells should also be performed in vaccinated individuals without previous SARS-CoV-2 infection.  Figure 1 of the supplementary files should be in the Material and Method section of the manuscript. 

The manuscript has several limitations besides the low number of samples, the possibility of reinfection should be stated as well as the possible responses to the vaccine in not infected individuals 

Minor typo errors were encountered

Author Response

Author Response to Reviewer 3

We thank the Reviewer for their favorable review of our manuscript entitled “Humoral Immune Responses in Patients with Severe COVID-19: A Comparative Study between individuals infected by SARS-CoV-2 during the Wild-type and the Delta periods”. Our manuscript has been revised according to the suggestions and concerns raised by the Reviewer.

Below, please find our point-by-point responses to the Reviewers’ comments. Additionally, we specify the corresponding line numbers where the text has been added or modified and highlight these changes in green in the revised submission.

Point #1: The article is interesting, although the low number of patients limited the study.

Response: We agree with the reviewer that the number of patients is relatively small. We have compensated for this shortcoming by conducting a large number of serological tests in which sera have been tested. However, we want to note that the presented samples are now already unique. At present, when as a result of the COVID-19 pandemic and a large vaccine campaign, many individuals have already been vaccinated and revaccinated, and had been repeatedly infected, it is rather problematic to find a sample in which patients were unvaccinated and infected only once. This is the sample we used.

We have included the following phrase at the end of the Discussion (Line 496):

The limitation of this study is the relatively small sample size.

Point #2: Figure 1 of the supplementary files should be in the Material and Method section of the manuscript.

Response: The reference to Supplementary Figure 1 (Study design) has been moved to the “Materials and Methods” section (Line 99).

Point #3: The possibility of reinfection should be stated as well as the possible responses to the vaccine in not infected individuals.

Response: Please see Lines 91-92, in which it is said that all patients have “no history of vaccination against SARS-CoV-2 and there were no reports of previous infection with COVID-19.”

If we consider the possibility of an asymptomatic infection before hospitalization, we can say the following. The likelihood of reinfection was extremely low during the D614G period. There is a small chance of reinfection during the Delta period. It is important to remember that the infection in the patients examined occurred very severely (hospitalization and intensive care unit) due to their likely immunocompromised state. It is difficult to accept that these patients could have an asymptomatic infection before the Delta period under these conditions.

Reviewer 4 Report

Since the emergence of the SARS-CoV-2 virus near the end of 2019, the virus, as expected, has continually evolved.  The high mutation rate of the virus has made it difficult to keep pace with the changes required in protective vaccines, as well as in passive prophylactic antibody treatments, to control the pandemic.  While it is acknowledged that the virus is quite adept at eluding strategies to control it, there have been few studies directly aimed at comparing the immune response raised against a particular variant of the virus for its efficacy against other variants.  In other words, can the neutralizing antibody responses to different variants be distinguished?  Also, can this information be useful in vaccine design?

In this study, a cohort of patients who acquired acute Covid-19 resulting from infections contracted during the circulation periods of either the D614G or Delta variants were evaluated for the specificity of their humoral response.  The authors conclude from the data that there are, indeed, two variant-specific responses to the two viruses; in particular, serum samples from the D614G period exhibited significantly stronger binding to homologous RBD and spike preparations than to those derived from the Delta variant.  These findings support the need to develop variant-specific vaccines and possible even therapeutics to prevent another pandemic surge.  The study is significantly strengthened by the development and usage of six different, complementary assays.

The major problem with the manuscript lies in the experimental design.  Specifically, the authors make it clear that the assignment of the patients to either of the D614G- or Delta-infected is never directly confirmed by genomic sequencing.  Although the authors contend that there is a high degree of certainty that this is, indeed, the case, the failure to confirm it is a glaring deficiency of the study.  If the entire study is based on the identity of the infecting virus, one would expect this to be confirmed rather than assumed, not matter how high the degree of certainty may be.

I am confused by the meaning of the term WT throughout the manuscript.  Is this the D614G virus?  Or do they use the term to mean the virus that is homologous to the serum that is being evaluated?  This needs to be clarified for the reader. Please specify in the text. 

In lines 404-406, the authors make the point that the use of variant-specific vaccines in difficult due to the complex, laborious and expensive nature of their development.  This is, of course, accurate.  But the authors may also want to extend this to the use and development of prophylactic antibody treatments, e.g. Evusheld, that are used to provide a level of protection for immunocompromised patients, who are unable to mount their own antibody responses to either the virus or vaccines.  Because large stocks of these antibodies can at any time become obsolete if the virus mutates to an extent that the antibodies do not neutralize it, this approach is losing its attraction for the industrial sector.  The authors may want to add this point as additional justification for this study.

English usage is fine.

Author Response

Author Response to Reviewer 4

We thank the Reviewer for their favorable review of our manuscript entitled “Humoral Immune Responses in Patients with Severe COVID-19: A Comparative Study between individuals infected by SARS-CoV-2 during the Wild-type and the Delta periods”. Our manuscript has been revised according to the suggestions and concerns raised by the Reviewer.

Below, please find our point-by-point responses to the Reviewers’ comments. Additionally, we specify the corresponding line numbers where the text has been added or modified and highlight these changes in green in the revised submission.

Point #1: Specifically, the authors make it clear that the assignment of the patients to either of the D614G- or Delta-infected is never directly confirmed by genomic sequencing.  Although the authors contend that there is a high degree of certainty that this is, indeed, the case, the failure to confirm it is a glaring deficiency of the study.  If the entire study is based on the identity of the infecting virus, one would expect this to be confirmed rather than assumed, not matter how high the degree of certainty may be.

Response: We agree with this comment of the reviewer.

In our support, we can cite a reference to the very close paper [9] (Mahalingam et al. Omicron Infection Increases IgG Binding to Spike Protein of Predecessor Variants. Journal of Medical Virology 2023, 95, e28419, doi:10.1002/jmv.28419; Impact Factor 2023: 20.693), which says: “As corresponding respiratory samples were not available to confirm infection, we inferred the infection type (Wild, Delta or Omicron and Delta and Omicron breakthrough among vaccinated) based on the circulation of variants over the study period by sequencing of respiratory samples in the same week as the corresponding sample.”

However, we have included the following phrase at the end of the Discussion (Lines 496-498):

The limitation of this study is the relatively small sample size. In addition, the as-signment of the patients to either D614G or Delta-infected was based on the period of dis-ease, but was not directly confirmed by genomic sequencing.

Point #2: I am confused by the meaning of the term WT throughout the manuscript. Is this the D614G virus?  Or do they use the term to mean the virus that is homologous to the serum that is being evaluated?  This needs to be clarified for the reader. Please specify in the text.

Response: Indeed, we did not use the terms that very clearly define the WT and D614G antigens.

Под термином WT мы подразумеваем Spike from the Wuhan-Hu-1 strain. D614G strain отличается от WT единственной мутацией в 614 положении. In ELISA, sVNA, and ADNKA assays we used WT RBD, который соответствует RBD from D614G virus. Для простоты этот RBD мы обозначали как WT. In mELISA, pVNA, and fcVNA assays мы использовали D614G Spike, что мы отразили в тексте и на соответствующих рисунках.

By WT we mean Spike of the Wuhan-Hu-1 strain. By WT we mean Spike of the Wuhan-Hu-1strain. The D614G strain is distinct from WT due to a single mutation in 614 position. In ELISA, sVNA, and ADNKA assays, we used WT RBD, which matched the RBD from the D614G virus. For simplicity, we have designated this RBD as WT. In mELISA, pVNA, and fcVNA assays we used D614G Spike, which we corrected in the text and in the corresponding figures.

For clarification, we included the following phrase:

Lines 104-105: The RBD from the Wuhan-Hu-1 strain was designated as wild-type (WT) RBD, and it matched the RBD from the D614G strain.

Point #3: In lines 404-406, the authors make the point that the use of variant-specific vaccines in difficult due to the complex, laborious and expensive nature of their development.  This is, of course, accurate.  But the authors may also want to extend this to the use and development of prophylactic antibody treatments, e.g. Evusheld, that are used to provide a level of protection for immunocompromised patients, who are unable to mount their own antibody responses to either the virus or vaccines.  Because large stocks of these antibodies can at any time become obsolete if the virus mutates to an extent that the antibodies do not neutralize it, this approach is losing its attraction for the industrial sector.  The authors may want to add this point as additional justification for this study.

Response: We thank the Reviewer for a very valuable hint. Indeed, in our Institute Evusheld is used to treat patients with common variable immunodeficiency. We have included the following phrase in the Discussion (Lines 411-417):

In addition to vaccination, protection against COVID-19 can be achieved by treating with monoclonal antibodies (mAbs) against SARS-CoV-2 antigens. The problem of increased immune escape of the new SARS-CoV-2 variants is common to both variant-specific vaccines and therapeutic mAbs [27], so the study of cross-reactivity of sera from recovered patients with different SARS-CoV-2 variants can also serve as a valuable resource for the development of novel prophylactic monoclonal antibodies (mAbs).

  1. Cox, M.; Peacock, T.P.; Harvey, W.T.; Hughes, J.; Wright, D.W.; COVID-19 Genomics UK (COG-UK) Consortium; Willett, B.J.; Thomson, E.; Gupta, R.K.; Peacock, S.J.; et al. SARS-CoV-2 variant evasion of monoclonal antibodies based on in vitro studies. Nat Rev Microbiol 2023, 21, 112–124, doi:10.1038/s41579-022-00809-7.

Reviewer 5 Report

I am not an expert in the field of immunology, but I am interested in reviewing this manuscript. Despite the humanity has experienced the spread and circulation of several SARS-CoV-2 variants, limited attention has been paid to direct comparison of immunity after COVID-19 caused by its different variants. Filatov and coworkers investigated and compared the different levels of humoral responses induced by natural infection with SARS-CoV-2 variants in Moscow population. It seemed the research design was appropriate, and the results were sound.

However, there are some concerns as following before I recommend it to be accepted:

1. The research focused on elderly patients, why not include young and middle-aged people? As revealed by the Title and Abstract, this study seemingly covered patients of all age groups.

2. All of the patients in this study were infected for the first time and without vaccination. As you see, nowadays people got infected by SARS-CoV-2 more easily than before. How about the humoral immune responses for the patients infected more than one time?

3. In this study, the patients were infected by D614G and Delta variants, which appeared two years ago. The virus mutates too quickly. Nowadays, Omicron variant are much more frequently encountered. As shown in Figure 1, Moscow population had been suffered the Omicron period in 2022. Why not include the patients infected by Omicron variant?

Minor editing of English language required.

Just a few typo errors, for instance:

1. P3L139: superscript ‘3.6 × 106 cells/100 mm’ → 3.6 × 106 cells/100 mm’

2. P4L186: superscript ‘5 × 104 cells/well’ → 5 × 104 cells/well’

3. P5L221: superscript ‘10 × 105 cells’ → 10 × 105 cells

4. P14: Page range for Ref. [11] was missing.

Author Response

Author Response to Reviewer 5

The favorable evaluation of our manuscript by Reviewer #5 is greatly appreciated and we are happy to receive such positive feedback.

Point #1: The research focused on elderly patients, why not include young and middle-aged people? As revealed by the Title and Abstract, this study seemingly covered patients of all age groups.

Response: We agree with this comment of the reviewer. Indeed, the study involved elderly patients. Only hospitalized patients were included in our study. This is because all hospitalized patients were well examined and characterized and such a sample it is logistically easier to collect. The hospital admitted only severe patients with COVID-19, who were generally in the age group. This explains some shift of our sample towards elderly patients. Despite the age shift, both samples (D614G and Delta periods) were overall similar in terms of age (Please see Line 265 and Supplementary Table 1)

Point #2: All of the patients in this study were infected for the first time and without vaccination. As you see, nowadays people got infected by SARS-CoV-2 more easily than before. How about the humoral immune responses for the patients infected more than one time?

Response: We agree with the reviewer that reinfection is a very important and interesting issue. To study it, it is necessary to collect a separate sample of multiple infected patients. Unfortunately, at the moment we do not have such a sample, but this study should definitely be provided for future work.

Point #3: In this study, the patients were infected by D614G and Delta variants, which appeared two years ago. The virus mutates too quickly. Nowadays, Omicron variant are much more frequently encountered. As shown in Figure 1, Moscow population had been suffered the Omicron period in 2022. Why not include the patients infected by Omicron variant?

Response: We agree with this excellent suggestion. Omicron period deserves separate consideration. Indeed, we tried to collect a sample of patients who were infected during the Omicron period, but this work is greatly complicated by the high heterogeneity of patients during this period. Various Omicron sub-variants (BA.1, BA.2, BA.4, BA.5, BA.2.12.1, BA.2.75, BQ, and XBB) often circulated simultaneously. For the correctness of such a study, it was necessary to carry out variant-specific sequencing. We anticipate that we will be able to perform such a study in the future.

Minor suggestions:

Point #1: P3L139: superscript ‘3.6 × 106 cells/100 mm’ → ‘3.6 × 106 cells/100 mm’.

Response: Corrected.

Point #2: P4L186: superscript ‘5 × 104 cells/well’ → ‘5 × 104 cells/well’.

Response: Corrected.

Point #3: P5L221: superscript ‘10 × 105 cells’ → ‘10 × 105 cells’

Response: Corrected.

Point #4: P14: Page range for Ref. [11] was missing.

Response: Corrected.

Round 2

Reviewer 3 Report

The manuscript has been partially improved; however, I am still concerned with the low number of samples and the analysis. The authors claim that reinfection was low, but only 26 individuals. There is also no record of infection of other pathogens, which, in this cohort, does not seem the case. The limitations of the study should be in a separate section. Again I would consider this only a pilot study and the authors should modify the title and the discussion accordingly

Minor errors in the text.

Reviewer 4 Report

In this revision of a previous submission, a comparison is made of the immune response to a particular variant of SARS-CoV-2 to a heterologous variant.  The authors use a complementary set of assays to determine the efficacy of serum samples derived from the period when the D614G period exhibited against the Delta variant.  The data show that the D614G variant sera demonstrate a significant degree of specificity for the homologous variant compared to the Delta variant.  These findings highlight the need to continually update vaccines as the virus evolves so that our defenses are optimal against the virus.

The major criticism of the previous version was that the identity of the D614G and Delta variants was not confirmed, but rather assumed as they were isolated during the period of time when each variant was predominant.  Although such direct confirmation cannot be obtained, the authors have added more direct statements asserting to this fact and stressing that the assumption is based solely the time of isolation coinciding with the prevailing variant circulating at the time.  A few other more minor points were also addressed.  The manuscript is now considered acceptable for publication.

MInor corrections required.

Author Response

We are grateful to the Reviewer for accepting our arguments about classifying patients as D614G- and Delta-infected by disease period, not by genomic sequencing.

The reviewer's comments will be taken into account in our future studies.